# Evaluation of Bi-Lateral Co-Infections and Antibiotic Resistance Rates among COVID-19 Patients

**DOI:** 10.3390/antibiotics11020276

**Published:** 2022-02-19

**Authors:** Naveed Ahmed, Madiha Khan, Waqas Saleem, Mohmed Isaqali Karobari, Roshan Noor Mohamed, Artak Heboyan, Ali A. Rabaan, Abbas Al Mutair, Saad Alhumaid, Salman A. Alsadiq, Ahmed S. Bueid, Eman Y. Santali, Jeehan H. Alestad

**Affiliations:** 1Department of Microbiology, Faculty of Life Sciences, University of Central Punjab, Lahore 54000, Pakistan; namalik288@gmail.com (N.A.); waqas.saleem@pkli.org.pk (W.S.); 2Center for Transdisciplinary Research (CFTR), Saveetha Dental College & Hospitals, Saveetha Institute of Medical and Technical Sciences University, Chennai 600077, Tamil Nadu, India; dr.isaq@gmail.com; 3Department of Restorative Dentistry & Endodontics, Faculty of Dentistry, University of Puthisastra, Phnom Penh 12211, Cambodia; 4Department of Pediatric Dentistry, Faculty of Dentistry, Taif University, P.O. Box 11099, Taif 21944, Saudi Arabia; roshan.noor@tudent.edu.sa; 5Department of Prosthodontics, Faculty of Stomatology, Yerevan State Medical University after Mkhitar Heratsi, Str. Koryun 2, Yerevan 0025, Armenia; heboyan.artak@gmail.com; 6Molecular Diagnostic Laboratory, Johns Hopkins Aramco Healthcare, Dhahran 31311, Saudi Arabia; 7College of Medicine, Alfaisal University, Riyadh 11533, Saudi Arabia; 8Department of Public Health and Nutrition, The University of Haripur, Haripur 22610, Pakistan; 9School of Nursing, Wollongong University, Wollongong, NSW 2522, Australia; abbas.almutair@almoosahospital.com.sa; 10Administration of Pharmaceutical Care, Al-Ahsa Health Cluster, Ministry of Health, Al-Ahsa 31982, Saudi Arabia; saalhumaid@moh.gov.sa; 11Home Health Care Department, Ras Tanurah Hospital, Ras Tanurah 32817, Saudi Arabia; salmanalsadiq@yahoo.com; 12Microbiology Laboratory, King Faisal General Hospital, Al-Ahsa 31982, Saudi Arabia; bueid123@hotmail.com; 13Department of Pharmaceutical Chemistry, College of Pharmacy, Taif University, P.O. Box 11099, Taif 21944, Saudi Arabia; eysantali@tu.edu.sa; 14Immunology and Infectious Microbiology, Glasgow G1 1XQ, UK; Jeehanalostad@gmail.com

**Keywords:** COVID-19, antimicrobial resistance, antimicrobial stewardship, co-infections, hospital-acquired infections, SARS-CoV-2

## Abstract

In addition to the pathogenesis of SARS-CoV-2, bacterial co-infection plays an essential role in the incidence and progression of SARS-CoV-2 infections by increasing the severity of infection, as well as increasing disease symptoms, death rate and antimicrobial resistance (AMR). The current study was conducted in a tertiary-care hospital in Lahore, Pakistan, among hospitalized COVID-19 patients to see the prevalence of bacterial co-infections and the AMR rates among different isolated bacteria. Clinical samples for the laboratory diagnosis were collected from 1165 hospitalized COVID-19 patients, of which 423 were found to be positive for various bacterial infections. Most of the isolated bacteria were Gram-negative rods (*n* = 366), followed by Gram-positive cocci (*n* = 57). A significant association (*p* < 0.05) was noted between the hospitalized COVID-19 patients and bacterial co-infections. *Staphylococcus aureus* (*S. aureus*) showed high resistance against tetracycline (61.7%), *Streptococcus pyogenes* against penicillin (100%), *E. coli* against Amp-clavulanic acid (88.72%), *Klebsiella pneumoniae* against ampicillin (100%), and *Pseudomonas aeruginosa* against ciprofloxacin (75.40%). *Acinetobacter baumannii* was 100% resistant to the majority of tested antibiotics. The prevalence of methicillin-resistant *S. aureus* (MRSA) was 14.7%. The topmost symptoms of >50% of COVID-19 patients were fever, fatigue, dyspnea and chest pain with a significant association (*p* < 0.05) in bacterial co-infected patients. The current study results showed a comparatively high prevalence of AMR, which may become a severe health-related issue in the future. Therefore, strict compliance of antibiotic usage and employment of antibiotic stewardship programs at every public or private institutional level are recommended.

## 1. Introduction

Coronavirus or SARS-CoV-2 (severe acute respiratory syndrome) belongs to the coronaviridae family and is an enveloped single-stranded positive-sense RNA virus. SARS-CoV-2 is a beta strain of the coronavirus family that causes severe acute respiratory syndrome. The virus responsible for coronavirus disease was first reported in December 2019 in Wuhan, China. Later, in March 2020, the World Health Organization (WHO) declared it a pandemic [1]. In Wuhan, it was first observed as an outbreak of atypical pneumonia. Due to this disease, many hospitalized patients’ illnesses became more serious, and COVID-19 cases doubled within a week. On further investigation, it was concluded that the coronavirus family’s beta strain was responsible for this infection, which was later identified as SARS-CoV-2 [2].

After studying the infection data from infected patients, many critical characteristics concerning the pathophysiology of SARS-CoV-2 were discovered. The most important contributors were secondary bacterial and fungal infections, as the majority of patients were infected with bacterial and fungal infections after acquiring SARS-CoV-2. In many patients, viral infection was seen to disrupt the macrophage model of activity by distorting the TLR4 and 5 pathways, which could be the reason for secondary bacterial infection by promoting bacterial attachments [3]. Furthermore, the virus may cause mucosal cell death, which compromises the pathogen purging mechanism and promotes bacterial adhesion [4]. Interactions between viruses and host cells may result in the formation of pro-inflammatory markers, such as TNF-α, a cytokine that can harm host cells and may lead to opportunistic bacterial infections. Viral infection may also activate INF1, which promotes the response of Toll-like receptors (TLR) to the ligand lipopolysaccharide [3].

The COVID-19 pandemic put a massive burden on the healthcare system, leading to changes in standard patient care approaches, increasing the incidences of hospital-acquired infections (HAIs). This additional COVID-19 burden, and long-term treatment of the associated patients, may have a significant association with HAIs [5]. The research community of AMR observed the transmission of SARS-CoV-2 and its interaction with other diseases worldwide, especially with secondary bacterial infections. It was concluded that there was a significant association between the usage of antibiotics and COVID-19 interventions that increased AMR rates [6]. COVID-19 patients receive antibiotic therapy for two reasons. The first one is COVID pneumonia symptoms with bacterial infection, and the second is a secondarily-acquired bacterial co-infection [7]. These reasons can determine the AMR evaluation in a specific population, the emergence of a pathogen, its transmission and infection burden [8].

Antibiotic stewardship programs and infection prevention and control practices may vary with locality. Empirical therapy is used for the primary treatment of critically ill patients in which a wide range of pathogens need to be covered, which may lead to the prescription of broad-spectrum antibiotics, like carbapenem and vancomycin. During the first phase of COVID-19, clinicians were faced with whether to use antibiotics as a treatment choice or not since they can have a significant impact on high AMR rates [9]. Antibiotic usage in the COVID-19 pandemic may change the AMR scenario to the worst situation ever. Keeping in view the current pandemic and threat of high AMR rates, the current study was conducted to determine the prevalence of bilateral bacterial co-infections in COVID-19 patients and the AMR patterns among each of the isolated bacteria.

## 2. Materials and Methods

### 2.1. Study Setting and Duration

This study was conducted at a tertiary care hospital in Lahore, Pakistan, collaborating with the Department of Microbiology, Faculty of Life Sciences (FoLS), University of Central Punjab (UCP), Lahore, Pakistan, from 16 October 2021 to 21 December 2021. All of the patients suffering from COVID-19 infection were admitted to different COVID-19 setting wards (medical ward, nephrology ward, gastroenterology ward, hepatology ward and general ward), and surgical intensive care units (SICUs) of the hospital were included in the study. The outpatient department (OPD) patients who were not suffering from COVID-19 infection were excluded from the study. The patients who were referred to any other hospital due to a treatment facility not being available in our hospital, and those who did not sign the consent form for the study, were also excluded from the study. Ethical approval was granted before starting the current study.

### 2.2. Collection of Patient Data

A total of 1165 COVID-19 patients were included in the current study. The demographical characteristics (age and sex) of studied patients were recorded from the patient medical record. The patients’ symptoms, baseline comorbidities, clinical characteristics and outcomes (discharge or death) were also recorded, apart from their demographic characteristics.

### 2.3. Collection of Clinical Samples

Keeping the testing facility available, each patient was proceeded for one type of sample only to include and study a more diverse group of patients. The clinical samples, including blood (*n* = 391), urine (*n* = 273), sputum (*n* = 123), throat swabs (*n* = 87), tracheal aspirate (*n* = 113), bronchoalveolar lavage (*n* = 63) and pus (*n* = 115) were collected under strict sterile conditions followed by standard operating procedures (SOPs) for COVID-19 infection. After collecting the samples, these were immediately transported to the Microbiology laboratory for further laboratory testing.

### 2.4. Isolation and Identification of Bacterial Isolates

After receiving the samples in the Microbiology laboratory, the samples were placed in the biosafety cabinet to proceed with culture inoculation. Before inoculating samples on different culture media plates, the samples (except urine) proceeded to smear preparation for Gram’s staining. After the preparation of smears on the sterilized glass slides, these samples were inoculated on MacConkey blood and chocolate agar as primary, secondary and tertiary streaking protocol. The urine samples were inoculated on cysteine electrolyte deficient (CLED) agar medium using the zig-zag streaking protocol. On each single agar plate, only one type of sample was inoculated. The blood culture vials (aerobic and anaerobic) were processed in the automated blood culture system (BectAlert 3D) for up to 7 days until declared unfavorable. After culture inoculation, the agar medium plates were incubated at 37 °C for 18 to 24 h, except for the sputum cultures, which were also incubated in the microaerophilic environment for certain specific organisms. After the first reading, if there was no growth present on the plates, they were again incubated for the next 24 h; at the second reading (48 h), the test was declared as “No growth” if there was still no growth.

For bacterial identification, first, the Gram’s staining slides were observed under the microscope’s 100X lens, and results were recorded on the tested sample file. After the incubation period, the agar medium plates were observed for the appearance of bacterial colonies and correlated with the Gram’s staining results. The bacteria were identified based on the morphology characteristics of colonies, Gram’s staining results and their biochemical profiling. The list of biochemical tests and their description is given in Table 1.

### 2.5. Antimicrobial Susceptibility Testing

Isolated organisms were tested for particular antibiotics as per the clinical laboratory standards (CLSI) recommendations. Both the Kirby Bauer (also known as the disc diffusion method) and serial dilution methods were used for antimicrobial activity testing of isolated strains. To validate the performance accuracy of the test, the American Type Culture Collection (ATCC) *E. coli* 25,922 strain was tested.

A bacterial colony was suspended in sterile normal saline with MacFarland standard, 0.5% for the Kirby Bauer disk diffusion method. The prepared suspension was then lawned on Muller Hinton (MH) agar plates, and the antibiotic discs were placed as per CLSI guidelines. After placing the discs, the agar plates were incubated for 24 h at 37 °C, then observed for the zone of inhibitions around the disks.

For the serial dilution method, the prepared MacFarland standard was added with gradually increasing antibiotic concentrations in 10 different tubes and incubated for 24 h at 37 °C. The results were checked based on the turbidity and reported as minimum inhibitory concentration (MIC).

### 2.6. Statistical Analysis

Data were recorded in Microsoft Excel and SPSS (Version 26.0). The association between the hospitalized COVID-19 patients with secondary bacterial infections and the appearance of symptoms in COVID-19 patients was compared with those infected with bi-lateral co-infections using the Chi-square test. A *p*-value < 0.05 was considered significant.

## 3. Results

### 3.1. Demographical Characteristics of Coronavirus Disease 2019 (COVID-19) Patients

Of the 1165 recruited patients (confirmed positive for COVID-19 by real-time PCR using nasopharyngeal swabs), the majority were male (*n* = 652), and the remaining (*n* = 513 were female). However, most of the patients co-infected with various bacterial infections were female (*n* = 224). Detailed information regarding the general characteristics of patients is given in Table 2.

### 3.2. Sample-Wise Positive Ratio of Bacterial Cultures

Each patient was processed for one sample type only to collect a more diverse group of samples and cover more patients. Most of the collected samples in the current study were blood followed by urine. Sputum, wound swabs, tracheal aspirate, throat swabs and bronchoalveolar lavage. A significant association (*p* < 0.05) was noted between the hospitalized COVID-19 patients and the bacterial co-infections. The positive ratios for bacterial cultures in different samples are shown in Table 3.

### 3.3. Isolation and Identification of Clinical Bacterial Isolates among COVID-19 Patients

Most of the isolated bacteria were Gram-negative rods (*n* = 366), followed by Gram-positive cocci (*n* = 57). Among the Gram-positive bacteria, *n* = 34 showed a positive reaction to the catalase biochemical test, which then proceeded to the coagulase test, and all showed positive reactions, which confirmed the identification of *Staphylococcus aureus*. The colony characteristics of *S. aureus* on the blood agar plate are shown in Appendix A. A total of 23 Gram-positive bacterial isolates showed adverse reactions for the catalase biochemical test, further proceeded with the *Streptococcus* grouping and identified as *Streptococcus pyogenes*. Among the Gram-negative rods, 234 were lactose fermenters, as shown in Appendix A, and 132 were non-lactose fermenters (Appendix A). In the lactose fermenters, 133 bacteria showed pink mucoid colonies and were indole positive (*Escherichia coli*), while the remaining 101 showed flat pink colonies and were indole negative, and therefore, proceeded further for citrate testing, which showed positive reactions (*Klebsiella pneumoniae*). The non-lactose fermenters first proceeded for the oxidase test, among which 61 isolates showed positive reactions (*Pseudomonas aeruginosa*). The *Enterobacteriaceae* were counter-confirmed for identification using API 20E strips, which showed the exact identification, as confirmed by the manual biochemical tests, as shown in Appendix A. The prevalence of each bacterium among the studied COVID-19 patients is shown in Figure 1.

### 3.4. Antibiotic Susceptibility Patterns of Individual Isolated Bacterial Isolates

A high prevalence of *S. aureus* resistance against tetracycline (61.7%) was noted, followed by gentamycin (50%), ciprofloxacin (47%), levofloxacin (47%), clindamycin (47%) and erythromycin (47%). The prevalence of the methicillin-resistant *S. aureus* (MRSA) strain was 14.7%. At the same time, no case of vancomycin-resistant *S. aureus* (VRSA) was found. The antibiotic resistance patterns of each isolated Gram-negative and Gram-positive bacteria are shown in Table 4 and Table 5, respectively.

The sensitivity pattern of *Pseudomonas aeruginosa* on the MH agar plate is shown in Appendix A. The MIC testing of colistin (CT) on the MH agar plate is shown in Appendix A. The MIC of vancomycin (VA) on the MH agar plate with the sensitivity pattern of VA (sensitive) and cefoxitin (sensitive) discs is shown in Appendix A. In comparison, the MIC of VA on the MH agar plate with the sensitivity pattern of VA (sensitive) and cefoxitin (resistant), oxacillin (resistant) and tetracycline (sensitive) discs are shown in Appendix A.

### 3.5. Appearance of Symptoms in COVID-19 Patients

At the time of admission to the hospital, patients faced different symptoms. The most common symptoms faced by critically ill patients were fever followed by fatigue, dyspnea, chest pain, sneezing, cough, sore throat, dizziness, headache and vomiting. Patients with different comorbidities faced more complicated symptoms. Fever followed by shortness of breath was the most commonly occurring onset symptom in ICU patients, as well as a cough that might be dry. These three symptoms were faced by more than 50% of patients admitted to the ICU. According to their immune system weakness and related comorbidities, rare symptoms were also prevalent in many patients. A significant correlation (*p* < 0.001) was found between the appearance of symptoms in COVID-19 patients and those who were infected with bi-lateral co-infections. The top 10 observed symptoms that appeared in COVID-19 patients and those co-infected with different bacterial infections are shown in Figure 2.

## 4. Discussion

Antimicrobial resistance (AMR) is a global health problem that poses a severe threat to treating a wide range of bacterial infections. The high AMR rates are a significant threat for patients admitted to different hospital wards, and patients admitted to intensive care units (ICUs) [10]. For public health and life sciences, finding strategies to combat the advancement of antibiotic resistance is a significant challenge. There has been a rapid increase in multidrug-resistant (MDR) pathogenic bacteria worldwide in the past few decades, especially in the current COVID-19 pandemic. More infections caused by MDR micro-organisms do not affect regular treatment, and the last choice of antibiotic may lose its effect [6]. Keeping in mind the current scenario of the pandemic situation, the study was conducted to determine the prevalence of bacterial infection and their antibiotic susceptibility patterns.

Nosocomial infections are serious issues for all patients admitted to the hospital, especially those admitted to the medical ICU [11]. Most patients admitted to the medical ICU are immunocompromised, putting them at higher risk of getting a nosocomial infection. Most infections in the medical ICU are catheter-related infections [12]. At present, 5 to 10% of patients entering emergency hospitals get at least one infection, and the risk has increased during the last decade [13]. The ICU accounts for 5 to 15% of hospital beds and 10 to 25% of medical expenses, equivalent to 1 to 2% of the clinical expense of the gross public product of the USA [14]. Nosocomial infections have a significant impact in the United States; more than 2 million people are admitted to ICU every year. Of these, 5 to 35% of patients admitted to the ICU get a nosocomial infection [15]. A previous study from Singapore showed that 14.8% of COVID-19 patients were co-infected with various nosocomial infections [16]. The results of the current study showed that COVID-19 patients admitted to the hospital were at high risk of getting co-infected with hospital-acquired infection. The most common causes of infection were Gram-negative bacteria. The main reason for these infections might be patients’ immune-suppressive status.

The infectious agent (SARS-CoV-2) that causes COVID-19 is exceptionally contagious, spreading mainly via droplets and close contact [17]. Several familial infection clusters have been recorded, and some of the verified individuals were infected in hospitals [18]. A recent study from Belgium has reported bacterial co-infection in 40.6% of the patients. The most common bacteria with multiple genome copies were *S. aureus*, *H. influenzae* and *Moraxella catarrhalis* [19]. In Italy, the prevalence of HAIs among COVID-19 patients was 56.73% [20]. However, a study conducted in Germany reported 34% of bacterial co-infections [21]. In the current study, 36% of studied patients were found to be co-infected with various bacterial infections.

The AMR also becomes an economic burden due to extensive usage of antibiotics, an extended stay in the hospital, expensive antibiotics and additional laboratory tests. Activities expected to control the issue of AMR are the responsibilities of local governments, medical care suppliers, specialists and the general population [22]. Patients admitted to ICUs have more chances of becoming resistant to drugs because of several factors, including starting empirical antibiotic treatment without doing the bacterial cultures and antimicrobial susceptibility testing, effective use of the invasive devices, and use of the drugs, which leads to a decrease in immunity and several nosocomial infections. Inappropriate antibiotic use and extensive use of broad-spectrum antibiotics are the most significant factors which lead to high AMR rates [23]. In our study, we have found that *S. aureus* has high resistance against tetracycline. *E. coli* and *Klebsiella pneumoniae* were highly resistant to amoxicillin, *Acinetobacter baumannii* to amikacin, while the *P. aeruginosa* was highly resistant to ciprofloxacin.

In recently published studies, it was observed that MDR bacterial prevalence has increased [24]. A retrospective study showed increased carbapenem resistance as almost more than double [25]. There were multiple underlying reasons for increased AMR rates, one of them being prolonged hospital stay. In a previous study in China, the incidences of *Acinetobacter baumannii*, *Klebsiella pneumonia* and *Stenotrophomonas maltophilia* were the same in both COVID and non-COVID patients [26]. It was shown that carbapenem resistance among *Acinetobacter baumannii* was 91.2% and 75% for *Klebsiella pneumoniae.* Another reason for increased AMR rates was the high usage of antibiotics, which was more than 99%, as compared to previous years [27]. Although COVID-19 is a pandemic, AMR could be an unnoticed pandemic, especially during this period, as described in a previous study where more than 30,000 deaths occurred because of AMR in Europe alone [27]. A recent study from the USA reported 44% more cases of MRSA [28]. A recent study from Pakistan [9] showed that the prevalence of MRSA was 20% in COVID-19 patients admitted to ICUs. However, in the current study, the prevalence of MRSA was 14%, with no reported cases of VRSA.

## 5. Study Limitations

A comparatively small number of patients were recruited because of the single study protocol. The patient’s data about the previous history of consuming antibiotics and the dosage of antibiotics could not be obtained because of ethical issues from the institution. Further large-scale and multi-institutional studies are recommended.

## 6. Conclusions

The current study reported a significant prevalence of bacterial co-infections among the hospitalized COVID-19 patients and a high AMR rate. The immune-suppressive status of these patients could play an essential role in acquiring different hospital-acquired infections. These high AMR rates threaten the loss of antibiotic effectiveness, especially in hospitalized patients. To monitor the compliance of antibiotic usage, periodic audits and antibiotic stewardship programs are required at the institutional level.

## Figures and Tables

**Figure 1 antibiotics-11-00276-f001:**
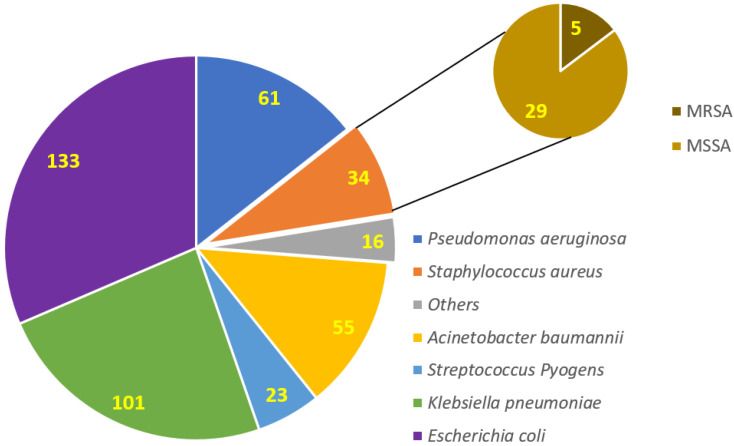
Prevalence of bacterial co-infections in COVID-19 patients. MRSA: methicillin-resistant *S. aureus*; MSSA: methicillin-sensitive *S. aureus*.

**Figure 2 antibiotics-11-00276-f002:**
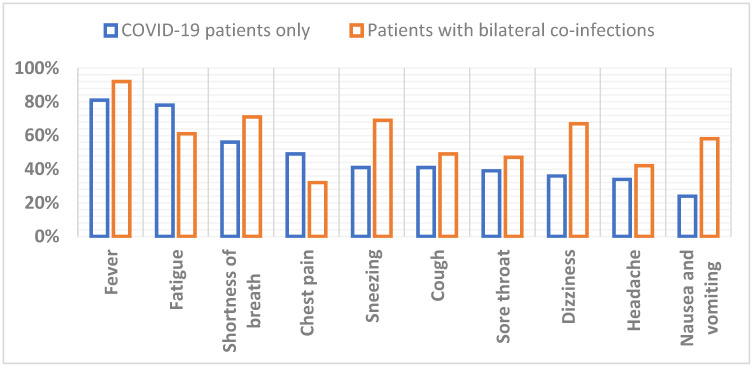
Top 10 symptoms observed in COVID-19 patients vs. COVID-19 patients with bilateral co-infections.

**Table 1 antibiotics-11-00276-t001:** List of biochemical tests and their description.

SR Number.	Test Name	Description
1	Catalase	Used to differentiate *Staphylococcus* spp. and *Streptococcus* spp.
2	Coagulase	Used to differentiate *Staphylococcus aureus* from others
3	Oxidase	Used to identify *Pseudomonas* spp.
4	Indole	Used to differentiate *E. coli* from other lactose fermenters
5	API	A strip of biochemical tests with reference database, different types were used to identify almost all kinds of organisms
6	DNAse	Used to identify *Staphylococcus aureus*

**Table 2 antibiotics-11-00276-t002:** General characteristics of patients included in the current study.

Characteristics		Number (*n*)	Percentage (%)
Gender	Male	652	55.96
	Female	513	44.03
Age (Years)	<30	152	13.04
	30–50	581	49.87
	>50	432	37.08
Comorbidities	Kidney diseases	152	13.04
	Hypertension	126	10.81
	Liver disease	114	9.78
	Hepatitis B	42	3.60
	Hepatitis C	86	7.38
	Meningoencephalitis	24	2.06
	Diabetes mellitus	97	8.32
	Gastrointestinal disorders	201	17.25
	None	323	27.72
Smokers	Yes	556	47.72
	No	609	52.27
Admission ward (COVID-19 unit)	Intensive care unit	197	16.90
	Gastro ward	186	15.96
	Nephrology ward	146	12.53
	Hepatology ward	236	20.25
	General medical ward	284	24.37
	Emergency	116	9.95

**Table 3 antibiotics-11-00276-t003:** Prevalence of positive bacterial cultures in different collected samples from COVID-19 patients.

Serial Number	Specimen	Frequency (*n* = 1165)	Positive for Bacterial Cultures (*n* = 423)
1	Blood	391	146
2	Urine	273	114
3	Sputum	123	56
4	Throat Swab	87	11
5	Tracheal Aspirate	113	41
6	Broncho alveolar lavage	63	23
7	Pus/Wound swab	115	32

**Table 4 antibiotics-11-00276-t004:** The antibiotic resistance patterns in Gram-positive bacteria.

Antibiotics	Resistance Percentage (%)	
	*Staphylococcus aureus* (*n* = 34)	*Streptococcus pyogenes* (*n* = 23)
Amikacin	11.76	NT
Chloramphenicol *	17.64	82.60
Cefoxitin	14.70	NT
Ciprofloxacin	47.05	NT
Co-trimoxazole	35.29	NT
Clindamycin	47.05	NT
Erythromycin *	47.05	60.86
Fusidic acid *	35.29	NT
Gentamicin	50.00	NT
Linezolid	0	NT
Penicillin	NT	100
Tetracycline	61.76	82.60
Teicoplanin	0	NT
Tobramycin	44.11	NT
Ceftriaxone	NT	82.60
Levofloxacin	47.05	82.60
Vancomycin	0	0

* Not reported in urinary isolates. NT: not tested.

**Table 5 antibiotics-11-00276-t005:** The antibiotic resistance patterns in Gram-negative bacteria.

Antibiotics	Resistance Percentage (%)			
	*E. coli* (*n* = 133)	*Klebsiella* spp. (*n* = 101)	*Acinetobacter baumannii* (*n* = 55)	*P. aeruginosa* (*n* = 61)
Ampicillin	84.21	100.00	NT	NT
Amp-clavulanic acid	88.72	90.09	NT	NT
Amikacin	12.03	16.83	100.00	13.11
Ceftriaxone	42.10	84.15	NT	NT
Cefuroxime	56.39	89.10	NT	NT
Cefixime	56.39	87.12	NT	NT
Ceftazidime	NT	NT	100	24.59
Chloramphenicol *	58.64	87.12	NT	NT
Ciprofloxacin	72.18	87.12	100	75.40
Levofloxacin	NT	NT	100	NT
Co-trimoxazole	81.20	73.26	100	NT
Gentamicin	38.34	27.72	96.36	19.67
Imipenem	6.01	16.83	92.72	27.86
Meropenem	6.76	16.83	92.72	29.50
Piperacillin-tazobactam	18.04	17.82	100.00	9.83
Tetracycline	85.71	70.29	100.00	NT
Tigecycline	0	0	74.54	NT
Tobramycin	51.87	0	90.90	29.50
Colistin	0	0	0	NT
Polymyxin B	0	0	0	NT
Cefepime	0	50.49	100.00	52.45
Nitrofurantoin **	13.53	42.57	NT	NT
Fosfomycin **	11.27	NT	NT	NT

* Not reported in urinary isolates. ** Only reported in urinary isolates. NT: not tested.

## Data Availability

The data will be shared upon reasonable request to the corresponding author.

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
