# Peer review of "Evaluation of Bi-Lateral Co-Infections and Antibiotic Resistance Rates among COVID-19 Patients"

_antibiotics, 2022, doi:10.3390/antibiotics11020276_

Round 1
Reviewer 1 Report
Dear authors,
The study is of current clinical importance and explores the association between COVID-19 infections and bacterial co-infections. It also looks at the impact of these infections on AMR.
The inclusion criteria have to be explicitly stated. There is no evidence presented that the antibiotic resistant strains colonise and cause infections in COVID patients more effectively than non-COVID patients. There were no control patients included nor was the previous use of antibiotics in these patients determined. The study populations is very small to draw meaningful conclusions. The authors have acknowledged these in their study limitations.
What is the correlation coefficient? You have used the term correlation but actually not done correlation analysis but instead a T-test.
Please see some specific comments below.
Line 34: disease symptoms
Line 39: Gram positive and Gram negative and not ‘gram’. check and correct through out the document.
Line:74: Do you mean ‘Toll-like receptors (TLR)’. Please correct the receptor name.
Line 78: ‘It was concluded that there was an association of behaviour and COVID 19 interventions that impact increasing antimicrobial resistance ‘lacks clarity. Behaviour ?
Line 89-91: ‘It was always a challenge for clinicians to about antimicrobial prescription during the phase of COVID 19, which may impact AMR, which depends upon the availability of local data.’ Revise sentence as is incomplete
Line 174: ‘the majority were male (n = 652) were male and the’ there is repetition, please correct it.
Best wishes
Author Response
Response to Reviewer 1 Comments
Dear authors,
The study is of current clinical importance and explores the association between COVID-19 infections and bacterial co-infections. It also looks at the impact of these infections on AMR.
Response: We would like to thank the academic editor and reviewers for taking their precious time to review this manuscript and give us comments. We would like to explicitly state that we agree with all the comments as these helped us improve the quality of our paper. We have made a conscious effort to answer all the remarks in the paper as advised by the reviewers and highlighted changes with red colour in the revised manuscript for their convenience. Changes have been carried out in the revised manuscript
- The inclusion criteria have to be explicitly stated. There is no evidence presented that the antibiotic resistant strains colonize and cause infections in COVID patients more effectively than non-COVID patients. There were no control patients included nor was the previous use of antibiotics in these patients determined. The study populations is very small to draw meaningful conclusions. The authors have acknowledged these in their study limitations.
Response: Thank you for your insightful suggestions and comments; antibiotics in COVID-19 patients may affect the AMR scenario in the future. Maybe there is no significance in the colonization of AMR bacterial strains in COVID-19 patients rather than in non-COVID patients, but our concern is that the way antibiotics have been used in this pandemic may create a big problem shortly. The irregular usage of antibiotics may lead to high AMR rates. The current study was conducted to evaluate the prevalence of bacterial co-infections in COVID-19 patients and to see their AMR patterns. We have reported a significant high AMR rate which may become a challenging situation later. The inclusion and exclusion criteria for our study has been mentioned in the manuscript, as we included only those patients who were critically admitted in different hospital wards. Those patients who were COVID-19 positive but not critically ill were excluded from the study. As we mentioned in the study limitations, the sample size was small because of the single centre study.
- What is the correlation coefficient? You have used the term correlation but actually not done correlation analysis but instead a T-test.
Response: Thank you for your insightful suggestions and comments; the association between the hospitalized COVID-19 patients with secondary bacterial infections and the appearance of symptoms in COVID-19 patients was compared with those infected with bi-lateral co-infections using the Chi-square test. A P < 0.05 was considered significant.
We have not proceeded with our data for Correlation > Bivariate analysis > Correlation coefficient (Pearson, Kendall's tau-b or spearmen) test because it was not the objective of our study. As we have mentioned about the limitations, we could not compare the results of those patients with the previous history. We have conducted an evaluation study rather than the correlation study in which we evaluated and analyzed the results statistically (Chi-square).
Please see some specific comments below.
- Line 34: disease symptoms
Response: Thank you for your insightful suggestions and comments; corrections has been carried out in the revised manuscript.
- Line 39: Gram-positive and Gram-negative and not 'gram'. check and correct through out the document.
Response: Thank you for your insightful suggestions and comments; corrections has been carried out in the revised manuscript.
- Line:74: Do you mean 'Toll-like receptors (TLR)'. Please correct the receptor name.
Response: Thank you for your insightful suggestions and comments; corrections has been carried out in the revised manuscript.
- Line 78: 'It was concluded that there was an association of behaviour and COVID 19 interventions that impact increasing antimicrobial resistance 'lacks clarity. Behaviour ?
Response: Thank you for your insightful suggestions and comments; the sentence has been rephrased as "It was concluded that there was a significant association between the usage of antibiotics and COVID-19 interventions that had an impact on increase AMR rates".
- Line 89-91: 'It was always a challenge for clinicians to about antimicrobial prescription during the phase of COVID 19, which may impact AMR, which depends upon the availability of local data.' Revise sentence as is incomplete
Response: Thank you for your insightful suggestions and comments; the sentence has been rephrased as "During the first phase of COVID-19, clinicians were faced with the question of whether to use antibiotics as a choice of treatment or not, since it may have a significant impact on high AMR rates. Antibiotics use in the COVID-19 pandemic may change the AMR scenario to the worst."
- Line 174: 'the majority were male (n = 652) were male and the' there is repetition, please correct it.
Response: Thank you for your insightful suggestions and comments; corrections have been carried out in the revised manuscript.

Reviewer 2 Report
This paper investigated the perveance of AMR in COVID-19 patients via biochemical characterization assays. The conclusion and data layout are somehow convincing, but still can be further strengthened by addressing the following concerns.
Major comments
- As it is a SARS-CoV-2-associated work, all samples were taken from COVID-19 patients, I am just wondering how these samples were treated to guarantee the biosafety because non-inactivated SARS-CoV-2 samples should be work in BSL-3 facility. Is it acceptable to handle SARS-CoV-2 samples in biosafety cabinets in Pakistan?
- Bacteria were only characterized by biochemical assays, molecular characterization is always necessary to confirm the results.
- How did the author different the co-infection bacteria and normal symbiotic bacteria? Especially in sputum, throat swabs, tracheal aspirate, and pus.
Minor comments
- Line 57, Wuhan is not the capital of China.
- Line 74, Toll-like receptors
- Line 82, co-infection
- Line 174, remove the repeated words ‘were male’
Author Response
Response to Reviewer 2 Comments
This paper investigated the perveance of AMR in COVID-19 patients via biochemical characterization assays. The conclusion and data layout are somehow convincing, but still can be further strengthened by addressing the following concerns.
Response: We would like to thank the academic editor and reviewers for taking their precious time to review this manuscript and give us comments. We would like to explicitly state that we agree with all the comments as these helped us improve the quality of our paper. We have made a conscious effort to answer all the remarks in the paper as advised by the reviewers and highlighted changes with red colour in the revised manuscript for their convenience. Changes have been carried out in the revised manuscript
Major comments
- As it is a SARS-CoV-2-associated work, all samples were taken from COVID-19 patients, I am just wondering how these samples were treated to guarantee the biosafety because non-inactivated SARS-CoV-2 samples should be work in BSL-3 facility. Is it acceptable to handle SARS-CoV-2 samples in biosafety cabinets in Pakistan?
Response: Thank you for your insightful suggestions and comments; the samples were collected from these patients by on-duty healthcare officers strictly following the SOPs as defined by WHO, CDC and NIH. After collecting the samples, these were transported to the microbiology laboratory in a 3-layer packaging sample transporting protocol. Once the samples had been reached in the laboratory, these were opened inside the biosafety cabinet by the trained staff following the complete SOPs for dealing with COVID-19 samples and then processed further accordingly.
As in Pakistan, there were no setups of BSL-3 laboratories at each hospital or diagnostic laboratory level; during the pandemic, the concerned health department authorities have planned and developed on an urgent basis BSL-3 like setups. In each of these setups, they have provided two BSL2 type A2 biosafety cabinets to process samples from COVID-19 patients. In the development of these setups, biosafety was kept in mind, and it was ensured that these should be sources for diagnosis rather than cross-transmission.
All of the samples were performed for diagnosis in following the SOPs as defined by WHO.
- Bacteria were only characterized by biochemical assays, molecular characterization is always necessary to confirm the results.
Response: We appreciate the molecular characterization of bacterial identification comment, but some of the techniques are newly discovered and cannot be used worldwide yet. The molecular characterization sometimes becomes very costly and not readily available at each institution level. Unfortunately, fully automated microbiological techniques are unavailable in Pakistan at each institution level. In the troubleshooting, these fully automated techniques need to be compared with gold standard methods of bacterial diagnosis.
In the current study, the bacteria were identified based on their biochemical reactions and colony morphology, Gram’s staining, and disk identification tests.
To confirm the final identification of Enterobacteriaceae, we have tested them using gold standard microbiological methods (cultures and biochemical), and Biomeurex API, which is considered as 100% sensitive and 96% specific; these percentages were calculated in a previously published study with a comparison of 16S RNA. Reference is given below.
(Nucera DM, Maddox CW, Hoien-Dalen P, Weigel RM. Comparison of API 20E and invA PCR for identification of Salmonella enterica isolates from swine production units. J Clin Microbiol. 2006 Sep;44(9):3388-90. doi: 10.1128/JCM.00972-06. PMID: 16954281; PMCID: PMC1594722.)
- How did the author different the co-infection bacteria and normal symbiotic bacteria? Especially in sputum, throat swabs, tracheal aspirate, and pus.
Response: Thank you for your insightful suggestions and comments; symbiotic bacteria present in the upper respiratory tract exceedingly abundantly; it is pretty challenging to differentiate either bacteria are normal flora or involved in the infection. Hence because the samples belonged to the immunocompromised patients, these were reported with the comments to rule out as per the sign and symptoms of the patient. Furthermore, bacterial co-infection was ruled out by a Direct smear of the upper respiratory tract as sputum gives a clue if direct smear showed more WBCs, predominantly neutrophils in most of the conditions.
Minor comments
- Line 57, Wuhan is not the capital of China.
Response: Thank you for your insightful suggestions and comments; corrections have been carried out in the revised manuscript.
- Line 74, Toll-like receptors
Response: Thank you for your insightful suggestions and comments; corrections have been carried out in the revised manuscript.
- Line 82, co-infection
Response: Thank you for your insightful suggestions and comments; corrections have been carried out in the revised manuscript.
- Line 174, remove the repeated words ‘were male’
Response: Thank you for your insightful suggestions and comments; corrections have been carried out in the revised manuscript.

Reviewer 3 Report
Thank you for the opportunity to review this manuscript.
The coronavirus disease 2019 (COVID-19) pandemic placed extraordinary demands on the healthcare system, resulting in modifications in routine patient care practices that have resulted the either increase risks for healthcare-associated infections and simultaneously, infection prevention and control practices became more visible in healthcare systems.
In general, the study gives an interesting multifactorial impression of the different aspects of the antibiotic resistance patterns among each of the isolated bacteria.
However, I suggest some improvements.
The introduction should better delve into the topic. I think that it is unfocused.
Line 58 – 75. I recommend to revisit this point emphasizing the relationship with long term care for COVID-19 and increasing risk of healthcare-associated infections. A good reference is Meghan A Baker, Kenneth E Sands, Susan S Huang, Ken Kleinman, Edward J Septimus, Neha Varma, Jackie Blanchard, Russell E Poland, Micaela H Coady, Deborah S Yokoe, Sarah Fraker, Allison Froman, Julia Moody, Laurel Goldin, Amanda Isaacs, Kacie Kleja, Kimberly M Korwek, John Stelling, Adam Clark, Richard Platt, Jonathan B Perlin, CDC Prevention Epicenters Program, The Impact of Coronavirus Disease 2019 (COVID-19) on Healthcare-Associated Infections, Clinical Infectious Diseases, 2021;, ciab688, https://doi.org/10.1093/cid/ciab688.
Methods are clear.
In the discussion it is appropriate to discuss of other studies that monitored this aspect.
Here are some references:
- Baccolini, V., Migliara, G., Isonne, C., Dorelli, B., Barone, L. C., Giannini, D., ... & Villari, P. (2021). The impact of the COVID-19 pandemic on healthcare-associated infections in intensive care unit patients: a retrospective cohort study. Antimicrobial Resistance & Infection Control, 10(1), 1-9.
- Baker, M. A., Sands, K., Huang, S. S., Kleinman, K., Septimus, E., Varma, N., ... & Perlin, J. B. (2021, November). 171. The Impact of COVID-19 on Healthcare-Associated Infections. In Open Forum Infectious Diseases (Vol. 8, No. Supplement_1, pp. S102-S103). US: Oxford University Press.
- Ong, C. C. H., Farhanah, S., Linn, K. Z., Tang, Y. W., Poon, C. Y., Lim, A. Y., ... & Marimuthu, K. (2021). Nosocomial infections among COVID-19 patients: an analysis of intensive care unit surveillance data. Antimicrobial Resistance & Infection Control, 10(1), 1-5.
Author Response
Response to Reviewer 3 Comments
Thank you for the opportunity to review this manuscript.
The coronavirus disease 2019 (COVID-19) pandemic placed extraordinary demands on the healthcare system, resulting in modifications in routine patient care practices that have resulted the either increase risks for healthcare-associated infections and simultaneously, infection prevention and control practices became more visible in healthcare systems.
In general, the study gives an interesting multifactorial impression of the different aspects of the antibiotic resistance patterns among each of the isolated bacteria.
Response: We would like to thank the academic editor and reviewers for taking their precious time to review this manuscript and give us comments. We would like to explicitly state that we agree with all the comments as these helped us improve the quality of our paper. We have made a conscious effort to answer all the remarks in the paper as advised by the reviewers and highlighted changes with red colour in the revised manuscript for their convenience. Changes have been carried out in the revised manuscript
However, I suggest some improvements.
The introduction should better delve into the topic. I think that it is unfocused.
Line 58 – 75. I recommend to revisit this point emphasizing the relationship with long term care for COVID-19 and increasing risk of healthcare-associated infections. A good reference is Meghan A Baker, Kenneth E Sands, Susan S Huang, Ken Kleinman, Edward J Septimus, Neha Varma, Jackie Blanchard, Russell E Poland, Micaela H Coady, Deborah S Yokoe, Sarah Fraker, Allison Froman, Julia Moody, Laurel Goldin, Amanda Isaacs, Kacie Kleja, Kimberly M Korwek, John Stelling, Adam Clark, Richard Platt, Jonathan B Perlin, CDC Prevention Epicenters Program, The Impact of Coronavirus Disease 2019 (COVID-19) on Healthcare-Associated Infections, Clinical Infectious Diseases, 2021;, ciab688, https://doi.org/10.1093/cid/ciab688.
Response: The introduction section has been revised according to the insightful suggestions and comments, and also the line 58-75 were modified according to the reference suggested in the comment. The reference has been added [5] in the revised version of manuscript.
“After studying the infection data from infected patients, many critical characteristics concerning the pathophysiology of SARS-Cov-2 were discovered. The most important contributors were secondary bacterial and fungal infections as majority of the patients were infected with bacterial and fungal infections after acquiring SARS-CoV-2. In many patients, viral infection was seen to disrupt the macrophage model of activity by distortion of TLR4 and 5 pathways, which could be the reason to secondary bacterial infection by promoting bacterial attachment [3]. Furthermore, the virus may cause mucosal cell death, which compromises the pathogen purging mechanism and promotes bacterial adhesion [4]. Interactions between viruses and host cells may result in the formation of pro-inflammatory markers such as TNF-α, a cytokine that can harm host cells and may lead to opportunistic bacterial infections. Viral infection may also activate INF1, which promotes the response of Toll-like receptors (TLR) to the ligand lipopolysaccharide [3]. The COVID-19 pandemic put a massive burden on the healthcare system, leading to changes in standard patient care approaches that might increases the incidences of hospital acquired infections (HAIs). This additional burden of COVID-19 and long-term treatment of these patients may have a significant association with HAIs [5].”
Methods are clear.
In the discussion it is appropriate to discuss of other studies that monitored this aspect.
Here are some references:
- Baccolini, V., Migliara, G., Isonne, C., Dorelli, B., Barone, L. C., Giannini, D., ... & Villari, P. (2021). The impact of the COVID-19 pandemic on healthcare-associated infections in intensive care unit patients: a retrospective cohort study. Antimicrobial Resistance & Infection Control, 10(1), 1-9.
Response: Thank you for your insightful suggestions and comments; corrections have been carried out in the revised manuscript. In Italy the prevalence of HAIs among COVID-19 patients was 56.73% [20].
- Baker, M. A., Sands, K., Huang, S. S., Kleinman, K., Septimus, E., Varma, N., ... & Perlin, J. B. (2021, November). 171. The Impact of COVID-19 on Healthcare-Associated Infections. In Open Forum Infectious Diseases (Vol. 8, No. Supplement_1, pp. S102-S103). US: Oxford University Press.
Response: Thank you for your insightful suggestions and comments; corrections have been carried out in the revised manuscript. A recent study from USA has reported 44% more cases of MRSA [28].
- Ong, C. C. H., Farhanah, S., Linn, K. Z., Tang, Y. W., Poon, C. Y., Lim, A. Y., ... & Marimuthu, K. (2021). Nosocomial infections among COVID-19 patients: an analysis of intensive care unit surveillance data. Antimicrobial Resistance & Infection Control, 10(1), 1-5.
Response: Thank you for your insightful suggestions and comments; corrections have been carried out in the revised manuscript. A previous study from Singapore has shown that 14.8% of COVID-19 patients were co-infected with various nosocomial infections [16].

Round 2
Reviewer 1 Report
My comments have been addressed by the authors.
Reviewer 2 Report
Satisfied revision.